# Adaption of an Evaporative Desert Cooler into a Liquid Desiccant Air Conditioner: Experimental and Numerical Analysis

**Mustafa Jaradat \*, Mohammad Al-Addous** and **Aiman Albatayneh**

Department of Energy Engineering; German Jordanian University, Amman Madaba Street, P.O. Box 35247, Amman 11180, Jordan; Mohammad.Addous@gju.edu.jo (M.A.-A.); aiman.albatayneh@gju.edu.jo (A.A.)
\* Correspondence: Mustafa.Jaradat@gju.edu.jo

**Abstract:** Desert coolers have attracted much attention as an alternative to mechanical air conditioning systems, as they are proving to be of lower initial cost and significantly lower operating cost. However, the uncontrolled increase in the moisture content of the supply air is still a great issue for indoor air quality and human thermal comfort concerns. This paper represents an experimental and numerical investigation of a modified desert air cooler into a liquid desiccant air conditioner (LDAC). An experimental setup was established to explore the supply air properties for an adapted commercial desert cooler. Several experiments were performed for air–water and air–desiccant as flow media, at several solutions to air mass ratios. Furthermore, the experimental results were compared with the result of a numerical simplified effectiveness model. The outcomes indicate a sharp reduction in the air humidity ratio by applying the desiccant solutions up to 5.57 g/kg and up to 4.15 g/kg, corresponding to dew point temperatures of 9.5 °C and 12.4 °C for LiCl and CaCl$_2$, respectively. Additionally, the experimental and the numerical results concurred having shown the same pattern, with a maximal deviation of about 18% within the experimental uncertainties.

**Keywords:** desert cooler; evaporative cooling; indoor air quality; liquid desiccant; effectiveness model; moisture removal

## 1. Introduction

Air conditioning represents a considerable part of current electrical power consumption due to the significant growth in populations, in addition to rapid developments in lifestyle and comfort standards [1,2]. In certain countries of the Middle East and North Africa (MENA), for example, Jordan, Lebanon, Syria, and Egypt, having air conditioners was considered as a luxury in the last few decades. However, the climate changes and hot summers in the last years have made it a must. The recent heatwaves have sharply increased the demand for air conditioners as simple ventilators were not enough to moderate the heat.

The total global electrical energy consumption for cooling in buildings using air conditioners accounts for about 20% nowadays [3]. Additionally, the share of energy consumption for space cooling is higher for regions with hot climates. In some of Middle East countries, such as Arab Gulf countries, the domestic ownership of air conditioning units is close to 100%.

There are various techniques for space cooling. The most utilized is the conventional vapor compression air conditioners. Although conventional air conditioners have many advantages, such as their reliability, stability, effective heat transfer rates, and compact sizes, they still have the problem of high overall electricity demand and peak electricity loads [4]. Besides, handling the latent load in vapor compression systems requires cooling the supply air beneath its dew point temperature.

Significant energy is required to reach the apparatus dew point (ADP) and further cooling below the ADP is required to compensate the bypass effect of the evaporator coils [5]. The dehumidified air must then be reheated to the required comfort level and this leads to extra energy waste [6]. Numerous issues have been associated with conventional coolers, starting with their relatively high costs, their operational costs, and maintenance costs; it has likewise been understood that recently the demand on using air conditioning for cooling has been increasing due to climate change and due to increase in the urban heat island impact, particularly in populated territories.

In some countries, because of the unaffordable conventional air conditioners (ACs) being accompanied by high operating expenses, people are now looking for alternatives of conventional power supply systems. Evaporative coolers have attracted much attention as an alternative to conventional air conditioning systems, as they are considered an economic solution to cool the supply air temperature [7]. Direct evaporative coolers use the latent heat of water to absorb heat from the supply air as it directly contacts the water and evaporates it. In this way, the system loses sensible heat with a lower dry bulb temperature while maintaining a constant wet bulb temperature. Simultaneously, the water vapor that evaporated is also added to the air, which increases the water content in the supply air leading to extreme indoor air quality (IAQ) issues and can increase the risk of respiratory diseases [8,9].

Regarding the most suitable climates where evaporative coolers are widely used, it has been noticed that they operate more efficiently in hot, dry regions; as the temperature of ambient air increases and its humidity decreases, the better will be the performance of the evaporative cooler at providing indoor thermal comfort. Those regions are the Middle East, Far East, some regions in North and West America, most regions in Australia, in addition to a number of European countries [10]. Figure 1 shows the regions that have hot and dry climates (red colored) that are preferable for desert coolers.

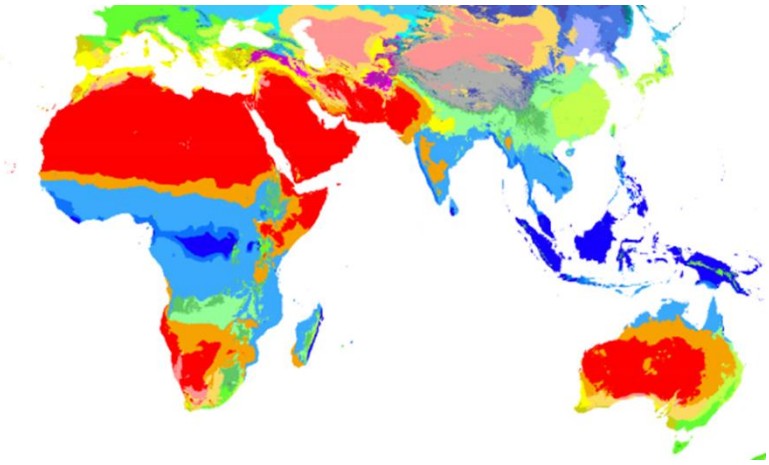

**Figure 1.** Köppen–Geiger climate classification map (1980–2016) [10].

Although evaporative cooling systems have been used as a more progressive and environmentally friendly replacement for conventional ACs, as it only requires electricity to drive a pump unlike vapor compression air conditioning systems which draw more electric energy [11–13], the issue remains with the fact that these systems rely on the ambient conditions. The cooling effectiveness of desert coolers is low comparing it to conventional vapor compression systems; they can also have relatively larger sizes and most importantly the potential for affecting human comfort is higher as it can increase the humidity level or even temperature may not be reduced enough. The increment in the water vapor level in the supplied air can lead to a fertile environment to sweat in, and for mold and bacterial growth. Furthermore, increased moisture levels can affect the stability of household hygroscopic material, such as woody furniture. Figure 2 shows the humidity levels been linked to comfort, mold growth, and the incidence of respiratory illness [14,15].

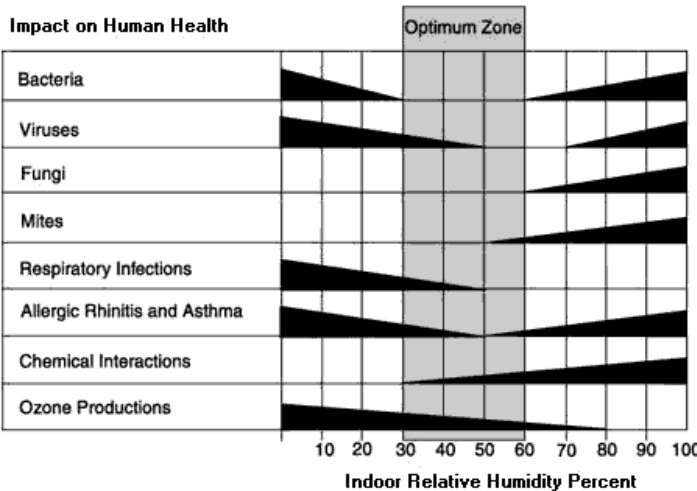

**Figure 2.** Recommended space humidity levels linked to its impact on human health [15].

Despite that evaporative cooling is known as an old system, there are few modifications in these systems, and very little regarding the humidity of the supply air leaving the cooler. Most of the modifications in the literature were coordinated toward the determination of air–water contact surfaces. Different cooling pads were evaluated regarding cooling efficiency, such as jute, luffa fibers, and palm fibers [16]; coarse and fine polyvinyl carbonate [17]; rice straw and palm leaf fibers [18]; and CELdek®, straw, and slice wood [19]. Additionally, several researchers have discussed the main factors when testing a cooling pad, such as surface area, thickness, and type and size of its perforations [20–22]. Some researchers also further discussed the decrease in electricity consumption by natural draught without the use of a blower [23].

One of the promising and energy efficient strategies for handling the latent load of air conditioning systems is to apply desiccant systems. By applying desiccants, air can be dehumidified without the need to cool the air below its dew point temperature [24]. The most used type of desiccant systems is solid desiccants and namely the desiccant wheels [25]. However, liquid desiccant systems possess the advantages of a separation between the absorption and the regeneration processes [26], regeneration at lower temperatures [27], and the possibility for energy storage in a thermochemical energy form [28–30]. Liquid desiccants are sorbents that have a high affinity to sorbates. In open cycle systems (under atmospheric pressure), the sorbate is the water vapor in the air.

The desiccant solution's effectiveness normally depends on the temperature and the concentration giving it a better performance at high concentrations and lower temperatures, so the controlling method for a solution can be attained by varying its temperature, concentration, or both according to the air specifications required for the output.

Due the appealing properties of desiccants with their affinity for more than water vapor, such as microorganisms, they are widely used in the applications in which air borne microorganisms could cause major issues, for example hospitals and medical facilities [31–33].

There are several common liquid desiccants that were investigated in many researches. Some examples of the investigated solutions are hygroscopic salt solutions, such as lithium bromide (LiBr), $CaCl_2$, and LiCl, or organic compounds like mono/triethylene glycol [34–36]. In the literature, LiCl and $CaCl_2$ solutions are the most investigated.

This paper represents an adaption of liquid desiccant solutions into a commercial desert cooler. $LiCl$-$H_2O$ with a mass fraction of 0.43 and $CaCl_2$-$H_2O$ with a mass fraction of 0.45 were used as desiccants. The desert cooler was firstly tested as a normal evaporative cooler and then it was tested with externally cooled desiccants with temperatures of 20 °C and 24 °C. The experiments for both desiccants were also performed by varying the solution mass flow rates at eight values, starting with

50 kg/h with an increment of 50 kg/h. Furthermore, the experimental results were compared with the numerical results of a basic effectiveness model.

## 2. Description of the Modified Desert Cooler

An experimental teat rig was constructed to experimentally study the air outlet characteristics. The analyses were performed on a typical commercial desert cooler that cost about US $80. The desert cooler represented a drip-type direct evaporative cooler with corrugated packing at the air outlet. Water streams flow down by gravity over the structured packing material and are recycled from the basin driven by a pump. The water sprinkled onto the top edges of the pad is distributed further by gravity and capillarity. Figure 3 shows the inner structure of the desert evaporative cooler and the setup as modified with desiccant solutions.

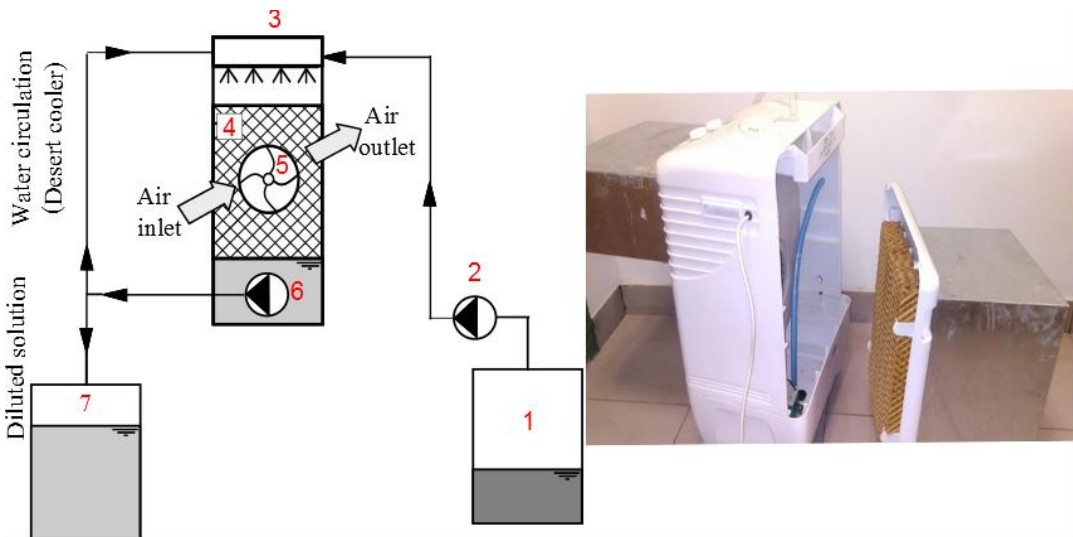

**Figure 3.** Inner configuration of the evaporative cooler and the setup with a desiccant solution (**left**), and the modified desert cooler (**right**). Numerical labels are (1) concentrated desiccant tank, (2) desiccant solution pump, (3) water/desiccant distributor, (4) packing material, (5) fan, (6) water circulation/desiccant-outlet pump, and (7) diluted desiccant tank.

The commercial desert cooler consisted of a corrugated packing material (4) made of cellulose with a volumetric surface area of 77 $m^2/m^3$. The cooler had a built-in pump (6) and a fan (5). The pump was originally used to circulate the water in the conventional evaporative cooler measurements. The built-in pump was used to extract the collected diluted solution to its tank (7).

The desert cooler inlet and outlet were modified by applying two rectangular ducts in order to connect it to the laboratory air ductwork, as shown in Figure 3 (right). The desiccant solution is introduced from above and it flows through a distributor above the cellulose packing material. The distributor is used originally to distribute the water in the desert evaporative cooler.

## 3. Instrumentation and Experimental Setup

Ambient air was supplied to an air handling unit accompanied by an air heater/cooler and humidifier/dehumidifier to create the target conditions required for each experiment. Air volume flow rate ($\dot{V}_a$) was measured using a volume flow meter. Air temperature ($\vartheta_a$) and relative humidity ($\varphi$) were measured using temperature and humidity sensors. Two temperature and relative humidity sensors were embedded at the inlet and outlet of the absorber. The three autonomous properties, namely temperature, relative humidity, and pressure, were used to derive all other air properties, such as the air humidity ratio and enthalpy, using humid air state equations from the American Society of Heating, Refrigerating and Air-Conditioning Engineers (ASHRAE) [37].

The desiccant solution circuit consisted of plastic tanks for the concentrated and diluted desiccant solutions. A pump was used to circulate the liquid desiccant. The volume flow rate of the desiccant solution was measured by a magnetic inductive flowmeter. The temperature of the desiccant solution was measured with Pt100 resistance thermometer sensors connected to the absorber inlet and outlet. The density of the solution ($\rho$) was measured by taking samples of the solution at the absorber inlet and outlet. The desiccant concentration was derived by the measured desiccant density and temperature by applying the correlations given by Conde [38]. Figure 4 shows the instrumentation setup of the absorber air circuit.

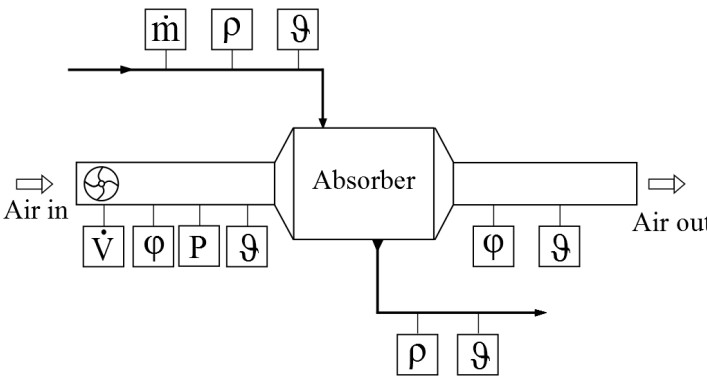

**Figure 4.** The experimental setup of the device tested with desiccant solutions. For the air circuit: volume flow rate ($\dot{V}$), relative humidity ($\varphi$), and temperature ($\vartheta$) were measured. For the desiccant solutions: mass flow rate ($\dot{m}$), density ($\rho$), and temperature ($\vartheta$) were measured.

The relevant parameters of the desiccant solution and air were measured at the inlet and outlet of the modified desert cooler. Table 1 shows the uncertainties of the connected instruments.

**Table 1.** Applied instrumentation and uncertainties as given by the manufacturer.

| Instrument | Model | Uncertainty | Range of Operation |
|---|---|---|---|
| Air temperature | Hygroflex 420 | 0.5 K | −50–100 °C |
| Relative humidity | Hygroflex 420 | 2% | 0%–100% |
| Air volume flow rate | Optiswirl 4070 DN 150 | 3% | 370–4800 $m^3$/h |
| Solution temperature | Pt100 | 0.5 K | |
| Solution volume flow rate | Optiflux 1050 | 1% | 20–3900 L/h |
| Solution density | L-Dens 323 | 1 $g/cm^3$ | 0.5–2 $g/cm^3$ |

The modified desert cooler was tested by varying the desiccant solution mass flow rate in the range of 50–400 kg/h in increments of 50. The air mass flow rate, temperature, and humidity ratio were kept steady as 400 kg/h, 32 °C, and 13.12 g/kg for the entire experiment. The mass fractions of the desiccant solutions were 0.43 for the LiCl-$H_2O$ solution and 0.45 kg/kg for the $CaCl_2$-$H_2O$ solution. The solution temperature varied at values 20 °C and 24 °C. Table 2 shows the experimental inlet parameters.

**Table 2.** Experimental setup of the modified desert cooler: inlet conditions.

| $\dot{m}_{sol}$ kg/h | $\dot{m}_a$ kg/h | $\vartheta_a$ °C | $\omega_a$ g/kg | $\vartheta_{sol}$ °C | $\xi_{LiCl}$ kg/kg | $\xi_{CaCl2}$ kg/kg |
|---|---|---|---|---|---|---|
| 50–400 | 400 | 32 | 13.12 | 20–24 | 0.43 | 0.45 |

A total of 32 experiments were carried out by varying one of the inlet parameters, denoted by the blue shaded cells in Table 2. The change in air temperature $\Delta\vartheta_a$, air humidity ratio $\Delta\omega$, and the rate of moisture removal $\dot{m}_v$ were assessed. The rate of moisture removal is given as shown in Equations (1) and (2) [31]:

$$\dot{m}_{v,AS} = \dot{m}_{da}(\omega_i - \omega_o), \tag{1}$$

$$\dot{m}_{v,SS} = \dot{m}_{salt}(X_o - X_i),\tag{2}$$

where $\dot{m}_{v,AS}$ represents the moisture removal rate calculated from the air side, which represents the change in the air humidity ratio $(\Delta\omega)$ multiplied by the dry air mass flow rate $(\dot{m}_{da})$; and $\dot{m}_{v,SS}$ represents the change in water content in the salt $(\Delta X)$ multiplied by the salt mass flow rate $(\dot{m}_{salt})$.

The water content per mass of salt $(X)$ is given in Equation (3):

$$X = \frac{1-\xi}{\xi}\tag{3}$$

where $\xi$ is the mass fraction of the salt solution, which represents the mass of the salt divided by the mass of the solution.

## 4. Numerical Model

A basic effectiveness model was modified for a quick estimation of the outlet conditions for the heat and mass exchanger from the entry conditions and the exchange surface. In addition, the model helps to determine the deviations between the measured data and simulation. This paper demonstrates a mathematical model for a cross-flow heat and mass exchanger for determining $\varepsilon$-NTU relations. The present model is based on the work of Stevens [39]. Effectiveness, $\varepsilon$, is defined as the ratio of the actual heat and mass transfer rate to the maximum possible rate, namely,

$$\varepsilon = \frac{\omega_{a,i} - \omega_{a,o}}{\omega_{a,i} - \omega_{eq}},\tag{4}$$

where $\omega_{eq}$ is the humidity ratio of air at equilibrium with the desiccant solution and is given as

$$\omega_e = 0.62198\frac{p_s\big(\vartheta_{sol,i}, \xi_{sol,i}\big)}{p - p_s\big(\vartheta_{sol,i}, \xi_{sol,i}\big)}.\tag{5}$$

The saturated pressure $p_s$ is realized as a nonlinear correlation of temperature and mass fraction according to [38] for aqueous solutions of LiCl and $CaCl_2$.

By applying the mass transfer coefficient $(\beta)$ and the area of exchanger surface $(A)$, the number of transfer units is derived:

$$NTU = \frac{\beta A}{\dot{m}_a}.\tag{6}$$

Applying the Lewis relationship, the rate of mass transfer coefficient to heat transfer coefficient is given in Equation (7) [40]:

$$\frac{\beta}{\alpha} = \frac{DLe^{1/3}}{\lambda}.\tag{7}$$

The $\varepsilon$-NTU relation for cross-flow configuration depends on the number of rows. For the corrugated packing material, with an infinite number of rows (air passages), the use of the approximate relation is described by Equation (8) and represents the infinite series solution taken from [39].

$$\epsilon = \frac{1}{C^*NTU}\sum_{n=0}^{\infty}\left\{\left[1 - e^{-NTU}\sum_{m=0}^{n}\frac{(NTU)^m}{m!}\right]\left[1 - e^{-C^*NTU}\sum_{m=0}^{n}\frac{(C^*NTU)^m}{m!}\right]\right\}.\tag{8}$$

## 5. Results and Discussion

### 5.1. Conventional Direct Evaporative Cooling

Prior to the desiccant investigations, the desert cooler was tested as a conventional evaporative cooler using water for the same air properties. Tap water was used with a temperature of 26 °C to fill

the desert cooler tank and the water was circulated internally. Table 3 shows the inlet parameters and outlet results of the desert evaporative cooler.

**Table 3.** Inlet and outlet conditions of the desert cooler tested in evaporative cooling mode.

| $\dot{m}_a$ kg/h | $\vartheta_{a,i}$ °C | $\omega_{a,i}$ g/kg | $\vartheta_{a,o}$ °C | $\omega_{a,o}$ g/kg | $\varepsilon$ − |
|---|---|---|---|---|---|
| 400 | 32.0 | 13.12 | 24.7 | 16.07 | 0.68 |

As shown in Table 3, a reduction in the air dry bulb ($\Delta\vartheta_a = 7.3\ K$) and an increment in the air humidity ratio ($\Delta\omega = 2.95\ g/kg$) were observed as a result of the heat and mass transfer between the air and water.

The performance of the desert cooler was assessed by determining the saturation effectiveness. The effectiveness of the desert cooler was $\varepsilon = 0.68$. The saturation effectiveness represents the extent to which the air exiting the desert cooler approaches the wet bulb temperature, as shown in Equation (9):

$$\varepsilon = \frac{\vartheta_{db,o} - \vartheta_{db,i}}{\vartheta_{db,o} - \vartheta_{wb}}. \tag{9}$$

*5.2. Modified Desert Cooler: Adiabatic Liquid Desiccant Dehumidification*

Two test sequences were performed to study the effect of solution flow rate and temperature on the absorption process. The test sequences were performed for the desiccant solutions of LiCl and CaCl$_2$. Even though that LiCl is more effective as a desiccant [32], CaCl$_2$ was also applied as it is more accessible and has a low cost compared to LiCl salt. The solution temperatures were set to 20 °C and 24 °C. The duration of each experiment was set to about 75 min. The shown experiments represent the averaged last 30 min of each experiment with a time-step of 10 s. This time represent a quasi steady-state density of the desiccant solutions at the absorber outlet.

The aim of the experiments was to study the supply air conditions represented by the rate of moisture removal and outlet temperature as shown in Figure 5, Figure 6, Figure 7, Figure 8, Figure 9, and Figure 10. Figures 5 and 6 show the rate of moisture removal rate from the airstream as a function of the solution mass flow rate using externally cooled desiccant solutions at 20 °C (Figure 5) and 24 °C (Figure 6). The experimental results of each of the solutions were compared; LiCl-H$_2$O (solid red line) and CaCl$_2$-H$_2$O (solid blue line). Moreover, the experimental results were compared with the results from the simplified effectiveness model (the dashed red and blue lines for LiCl and CaCl$_2$ solutions, respectively).

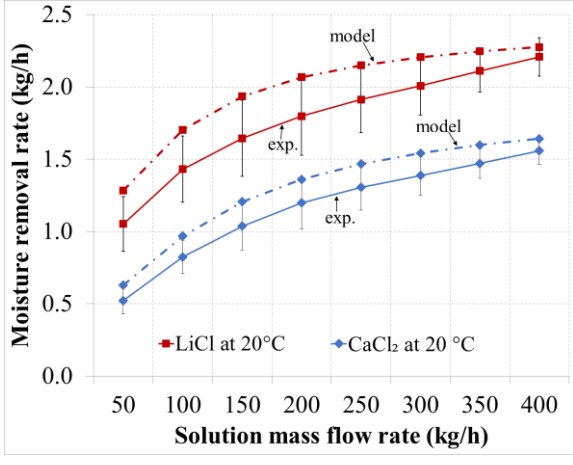

**Figure 5.** Absorbed water vapor versus the solution flow rate for the desiccants with an inlet temperature of 20 °C.

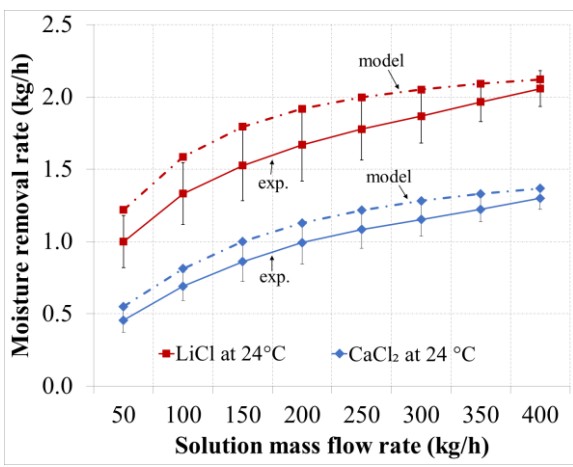

**Figure 6.** Absorbed water vapor versus the solution flow rate for the desiccants with an inlet temperature of 24 °C.

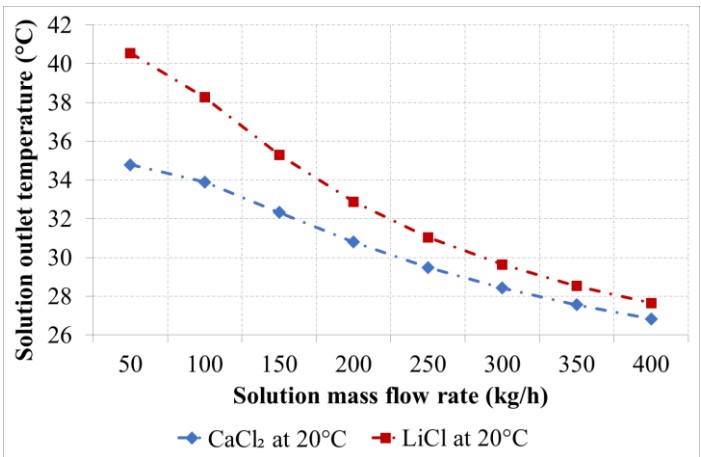

**Figure 7.** Solutions outlet temperature as a function of the solution flow rate for the desiccants with an inlet temperature of 20 °C.

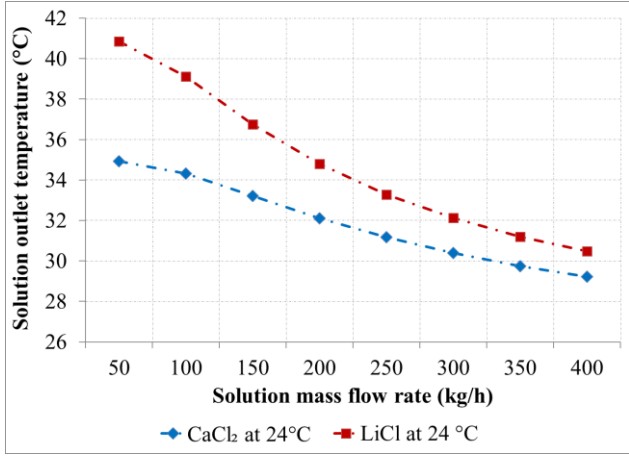

**Figure 8.** Solutions outlet temperature as a function of the solution flow rate for the desiccants with an inlet temperature of 24 °C.

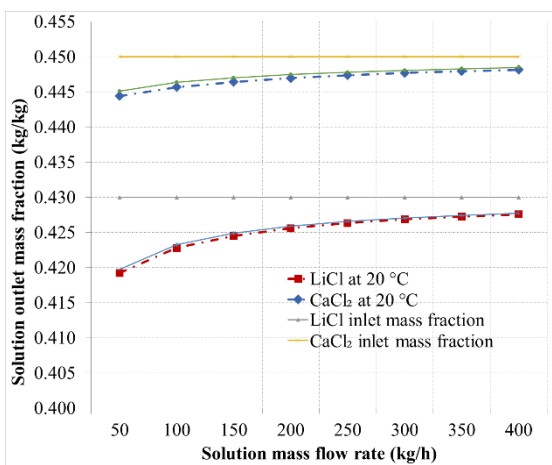

**Figure 9.** Mass fraction spread for the LiCl and CaCl₂ desiccant solutions at 20 °C and 24 °C as a function of solution flow rate.

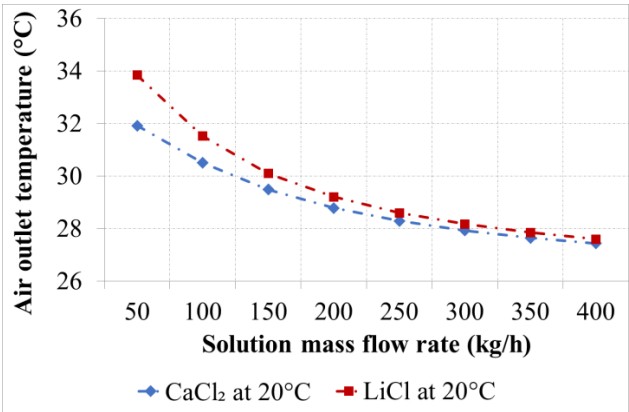

**Figure 10.** Air outlet temperature as a function of the solution flow rate for the desiccants with an inlet temperature of 20 °C.

As shown in Figures 5 and 6, the moisture removal rate increases by increasing the solution mass flow rate. For the LiCl solution there was an increase in the moisture removal rate of 1.16 kg/h (110% increase) for the given range of solution flow rate. This trend is expected as increasing the solution flow rate will lead to lower increments in the solution temperature, as shown in Figures 7 and 8. Increasing the solution flow rate also leads to lower decrement in the solution concentration, as shown in Figure 9. As a result, this will keep the solution at a low vapor pressure and thus improve the mass transfer of water vapor from air to the concentrated solution. Furthermore, an increased solution flow rate will enhance the wetted transfer area of the corrugated packing material.

The deviation between the experimental and numerical results was in the range of 3%–18% and decreased by increasing the solution flow rate. An explanation could be that increasing the solution flow rate will increase the surface wetting of the corrugated material and thus coming closer to the assumption of complete surface wetting assumed in the model. The deviation was within the experimental uncertainties of 20%.

The rate of moisture removal using the LiCl solution is higher than for the CaCl₂ solution due to its higher absorption capacity. The rate has also been noted to be higher for solutions with lower temperatures, hence at 20 °C. For the LiCl solution with an inlet temperature of 20 °C, the moisture removal rate was in the range of 1.05–2.21 kg/h compared to 0.52–1.56 kg/h for CaCl₂ for the same inlet temperature.

An increment in the solutions outlet temperature was observed as a result of the enthalpy of condensation and enthalpy of dilution. This increment is inversely proportional to the solution flow rate and is lower for the $CaCl_2$ solution compared to LiCl solution, as shown in Figures 7 and 8. An increment in the solution temperature of $\Delta\vartheta_{sol} = 20.6\ K$ was observed for the LiCl solution with an inlet temperature 20 °C and a mass flow rate of 50 kg/h compared to $\Delta\vartheta_{sol} = 14.8\ K$ for the $CaCl_2$ solution at the same inlet conditions. The increase in the solution temperature decreases with increasing the solution flow rate as a result of the low heat capacity of the desiccant solutions and the reduced exposure time. An increment of the solution temperature of about 7 K was observed for a solution flow rate of 400 kg/h for both desiccants.

Figure 9 shows the outlet mass fraction of both desiccants as a function of solution flow rate. The spread of desiccant solutions is inversely proportional to the solution mass flow rate, as shown in Figure 9. The spread of the mass fraction was higher for the LiCl solution with a maximal spread of 2.5% compared to the $CaCl_2$ solution with a maximal spread of 1.2%; both for a solution mass flow rate of 50 kg/h. While, the mass fraction spread was in the range of 0.6% and 0.4% for a solutions mass flow of 400 kg/h for LiCl and $CaCl_2$, respectively. Furthermore, the mass fraction spread was slightly affected by the solution inlet temperature for the given temperature range.

Figures 10 and 11 show the air outlet temperature as a function of solution flow rate for a solution inlet temperature of 20 °C and 24 °C. As an adiabatic process, the air outlet temperature decreased by increasing the solution flow rate. An increment in the air outlet temperature was observed with LiCl as the desiccant for a solution flow rate of 50 kg/h.

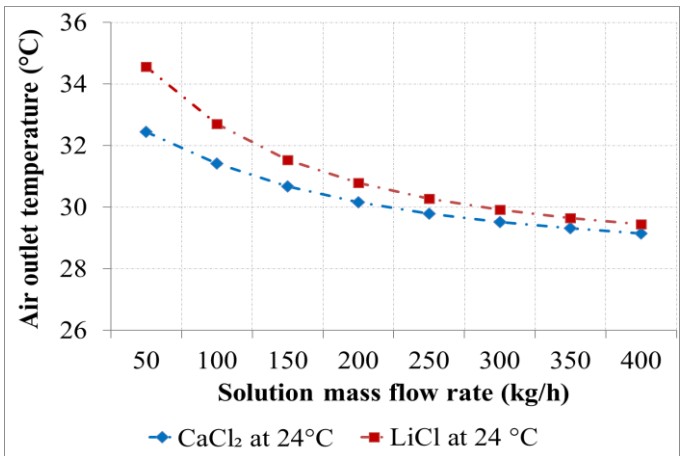

**Figure 11.** Air outlet temperature as a function of the solution flow rate for the desiccants with an inlet temperature of 24 °C.

A summary of the performed experiments is presented in Figure 12. The prototype was tested in three modes: in the conventional direct evaporative cooling mode (the red line), and adiabatic dehumidification using LiCl (blue points) and $CaCl_2$ (green points) solutions.

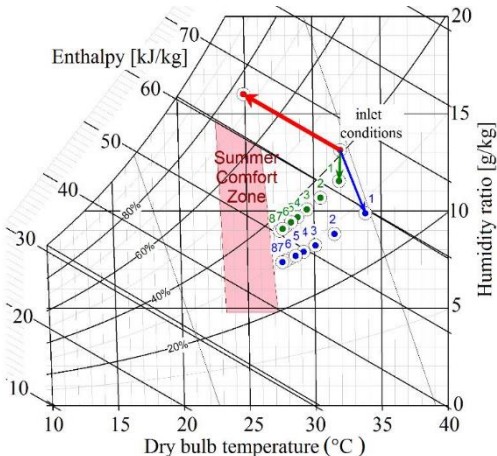

**Figure 12.** Illustration of the performed experiments on the psychrometric chart (Mollier diagram): The red line represents the conventional direct evaporative desert cooler, the green dots represents the performed experiments with CaCl$_2$ as a desiccant, and the blue dots represents the performed experiments with LiCl as a desiccant.

Figure 12 represents a graphical user interface for thermal comfort prediction based on ASHRAE Standard 55 [41]. Summer comfort zone specifies boundaries of air temperature and humidity for people in typical summer clothing during primarily sedentary activity. Since people feel comfort subjectively and each person has their own preferences for room climate, there is a comfort range of room air temperature and relative humidity in which the vast majority of room users feel comfortable. The limits of summer comfort zone, of ASHRAE Standard 55, were developed theoretically. As shown in Figure 12, this zone extends from around 23 °C to 27 °C and 35% to 75% relative humidity; i.e., the absolute humidity should be between 5 and 12 g/kg air for comfort.

The air that left the conventional desert cooler had a significant drop in the dry bulb temperature of 7.3 K. However, the air humidity ratio is significantly increased to 16.1 g/kg; this can be noticed while observing the red line path on the psychrometric chart (Mollier diagram) above. The air outlet conditions for the liquid desiccant system are shown in Figure 12, and the points were determined based on the experimental evaluation carried out in the paper. The results of absorption using LiCl and CaCl$_2$ solutions at 20 °C were chosen to be represented in the psychrometric chart. The blue line shows the path of the experimental results using the LiCl solution for the experiments 1 to 8 (by increasing the solution flow rate) while the green dots represent the experimental results using CaCl$_2$. As shown, the humidity ratio decreases significantly by applying the desiccant solutions and it is higher for the LiCl solution compared to the CaCl$_2$ solution. Additionally, the temperature does not decrease like it does in a conventional desert cooler, the air outlet conditions hit the comfort zone margins for a solution to air mass ratio of $r_m = 1$.

*5.3. Dehumidification Load*

For the dehumidification and cooling by applying liquid desiccants in the modified desert cooler, the cooling load is calculated by multiplying the air mass flow rate by the change in specific enthalpy as given in Equation (10) [5]:

$$\dot{Q}_{deh} = \dot{m}_{da}(h_i - h_o). \tag{10}$$

The specific enthalpy ($h$) of the air entering the modified evaporative cooler is $\left(h_i = 65.7 \frac{kJ}{kg}\right)$. For the performed experiments, presented in Figure 12, the dehumidification/cooling load ($Q_{deh}$) and the air outlet conditions are given in Table 4.

**Table 4.** Air outlet conditions and the dehumidification load for both desiccants at a temperature of 20 °C.

| Exp. | Air Outlet Conditions Using LiCl-$H_2O$ as Desiccant | | | | Air Outlet Conditions Using CaCl$_2$-$H_2O$ as Desiccant | | | |
|---|---|---|---|---|---|---|---|---|
| | $\vartheta_o$, °C | $\omega_o$, $\frac{g}{kg}$ | $h_o$, $\frac{kJ}{kg}$ | $\dot{Q}_{deh}$, kW | $\vartheta_o$, °C | $\omega_o$, $\frac{g}{kg}$ | $h_o$, $\frac{kJ}{kg}$ | $\dot{Q}_{deh}$, kW |
| 1 | 33.8 | 9.87 | 59.4 | 0.71 | 31.9 | 11.53 | 61.6 | 0.46 |
| 2 | 31.5 | 8.81 | 54.3 | 1.27 | 30.5 | 10.67 | 58.0 | 0.86 |
| 3 | 30.1 | 8.22 | 51.3 | 1.60 | 29.5 | 10.07 | 55.4 | 1.15 |
| 4 | 29.2 | 7.89 | 49.5 | 1.80 | 28.8 | 9.67 | 53.7 | 1.34 |
| 5 | 28.6 | 7.68 | 48.4 | 1.93 | 28.3 | 9.41 | 52.5 | 1.47 |
| 6 | 28.2 | 7.54 | 47.6 | 2.02 | 27.9 | 9.22 | 51.6 | 1.57 |
| 7 | 27.9 | 7.43 | 47.0 | 2.08 | 27.6 | 9.07 | 51.0 | 1.64 |
| 8 | 27.6 | 7.36 | 46.6 | 2.13 | 27.4 | 8.97 | 50.5 | 1.70 |

As shown in Table 4, the dehumidification load increases by increasing the solution flow rate up to 2.13 kW and 1.7 kW for solution mass flow rates of 400 kg/h of LiCl-$H_2O$ and CaCl$_2$-$H_2O$, respectively. The table also shows that the LiCl-$H_2O$ is more efficient in dehumidification compared to CaCl$_2$-$H_2O$, as expected. The percentage change in the dehumidification load was up to 25.3% for an air to solution mass ratio of $r_m = 1$.

However, an additional load is required for the regeneration of the diluted liquid desiccant solution that can be regenerated by solar energy.

## 6. Conclusions

In this paper, a modification of a direct evaporative (desert) cooler was performed by applying LiCl-$H_2O$ and CaCl$_2$-$H_2O$ as desiccants. The prototype was tested by running it in a direct evaporative cooler and in an externally cooled (adiabatic) desiccant system. Several experiments were performed by varying the solution flow rate at the two inlet temperatures. The experimental results were also compared with a single-node effectiveness model.

The results show more beneficial air outlet conditions by using liquid desiccants regarding the air humidity ratio: A reduction in the air humidity ratio of up to 5.76 g/kg for the LiCl solution and up to 4.15 g/kg for the CaCl$_2$ solution compared to an increase of 2.95 g/kg in the humidity ratio for the conventional desert cooler. Thus, the indoor air quality is significantly improved in the modified desert cooler.

The moisture removal rate calculated from the measured data was compared with the numerical results from the $\epsilon - NTU$ effectiveness model. The results show a systematic pattern and the results were in good agreement. The maximal deviation in the rate of moisture removal between the experimental and the modeled results was in the range of 18%.

The results show that the LiCl solution is more effective in moisture removal than the CaCl$_2$ solution. However, the difference in moisture removal between LiCl and CaCl$_2$ solutions decreased by increasing the solution flow rate. The percentage difference was 51% for $r_m = 0.125$ and it was reduced to 28% for $r_m = 1$. Considering the huge price difference between LiCl and CaCl$_2$, it is highly recommended to apply CaCl$_2$ to the conventional desert coolers, which will make the price competitive for the target group of users.

**Author Contributions:** Conceptualization, M.J.; methodology, M.J. and A.A.; software, M.J.; formal analysis, M.A.-A. and A.A.; investigation, M.A.-A.; resources, A.A.; data curation, M.A.-A.; writing—original draft, M.J.; writing—review and editing, A.A.; supervision, A.A. All authors have read and agreed to the published version of the manuscript.

**Funding:** This research received no external funding.

**Acknowledgments:** The authors are grateful for the support of the Deanship of Graduate Studies and Research at the German Jordanian University.

**Conflicts of Interest:** The authors declare no conflict of interest.

## Nomenclature

| Symbol | Meaning | Unit |
|--------|---------|------|
| $A$ | Area | $\text{m}^2$ |
| $\alpha$ | heat transfer coefficient | $\text{W/m}^2\text{.K}$ |
| $\beta$ | mass transfer coefficient | $\text{m/s}$ |
| $\varepsilon$ | effectiveness | |
| $h_{fg}$ | enthalpy of vaporization | $\text{kJ/kg}$ |
| $\dot{m}$ | mass flow rate | $\text{kg/h}$ |
| $p$ | pressure | $\text{Pa}$ |
| $r_m$ | air to solution mass flow ratio | |
| $X$ | mass of water per mass of salt | $\text{kg}_{\text{H2O}}/\text{kg}_{\text{salt}}$ |
| $V$ | volume | $\text{m}^3$ |
| $\Delta$ | difference | |
| $\lambda$ | thermal conductivity | $\text{W/m.K}$ |
| $\omega$ | humidity ratio | $\text{g}_{\text{w}}/\text{kg}_{\text{da}}$ |
| $\varphi$ | relative humidity | |
| $\rho$ | density | $\text{kg/m}^3$ |
| $\vartheta$ | temperature | $^\circ\text{C}$ |
| $\xi$ | mass fraction of the desiccant | $kg_{salt}/kg_{sol}$ |
| **Subscripts** | | |
| a | air | |
| $\text{CaCl}_2$ | calcium chloride | |
| i | inlet conditions | |
| da | dry air | |
| deh | dehumidification | |
| LiCl | lithium chloride as salt | |
| NTU | number of transfer units | |
| o | outlet conditions | |
| s | saturated | |
| sol | solution | |
| v | water vapor | |
| w | water | |

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
