# Peer review of "Adaption of an Evaporative Desert Cooler into a Liquid Desiccant Air Conditioner: Experimental and Numerical Analysis"

_atmosphere, doi:10.3390/atmos11010040_

Round 1

Reviewer 1 Report

Dear Authors,

Thank You very much for ths interesting paper. It is generally well written and clear. It is very practical what I consider as a advantage. I reccomend it to  be published nevertheless I have few suggestions/remarks:

I found single misspellings (for example p.1 l.19) I suggest to refer to specification of the meters/sensors You used or at least provide models I suggest to add citation of Lewis relationship Figures 5 and 6 are to small – it is hard to distinguish which is model which experiment I suggest to include an information about the Energy consumption or COP of the device – the increase of dessication is achieved at the expense of increased flow rate (probably also pressure drop) of solution. Such information can be then multiply cited. If I understood correctly You combined to desert coolers (one modified to be dessicant ). Have You investigated the concentration of solution in the outlet air? Is it harmful?

Once again thank You very much.

Author Response

Dear respectful reviewer,

On behalf of the co-authors and me I would like to thank you for the valuable comments. We have taken them into account, as far as possible, point by point hoping that the paper will now fulfil the requirements for publication in Atmosphere.

Reviewer 2 Report

Dear Authors,

Thank you for your manuscript about the adaption of an evaporative desert cooler into a liquid desiccant air conditioner.

There are some points for consideration left. Please take them into account and improve the decription of your experimental setup and mention the functional range incl. humidity ratio and other hygrometric quantities at the outlet:

Abstract

line 13 and ff: liquid desiccant air-conditioner and mention at least once the common abbreviation LDAC

line 18: may you support the humidity mixing ratio with dew point or another hygrometric quantity

Text

Figure 2: Have you determined the data/figure on your on? If not, please give a reference in the caption like you did it for Fig.1.

Figure 4: Improve the figure resolution and define every variable, - in the text or in the figure capiton or in table 1.

Table 1: May you list every quantity of your setup.
accuracy must be replaced with uncertainty
Which quantity has the enhancementfactor  k?
Is the density of a 0.43 aqueous LiCl-solution really 1 kg/m³?

in General for table 1
each uncertainty quantity without ±
RH is redundant for Air relative humidity
rephrase "air relative humidity" to "relative humidity" - Do you mean the generated relative humidity?

What is the total uncertainty of your modified setup? Please calculate the combined measurement uncertainty of your whole process with an enhancement factor, e. g. k = 2.

in general notation/nomenclature: Please check and use the quantity notation according to ASHREA's handout. http://www.ce.utexas.edu/prof/Novoselac/classes/ARE383/Handouts/F01_06SI.pdf

line 166: please order the temperature in an increasing way, 20 °C to 24 °C

line 172: put the colon after [30]

line 265: There is no Figure 98 in your manuscript. Please improve.

Figure 9:

In my opinion you also describe the recovery of the mass fraction. May be it is better to describe generated temperatures and humidity Levels over solution mass flow rate.

Fig 10 and 11:
As I read the y-axis description "supply air temp" vs x-axis "solution mass flow"
May be it is better if you show the air outlet temp as a function of the solution flow, as written in the caption.

Figure 12:

In this figure you point out the dry bulb temp. vs. humidity Ration in g/kg. It was quite difficult for me to find the wrap up of your data from the previous figures. May you Support this figure with a data table of obtained dry bulb temp and humidity ratios. Give also an information about outdoor air conditions next to your presented data and explain the summer comfort zone. Is there a reason why you did not hit values inside the summer comfort zone? Put "Mollier chart" into brackets after psychrometric chart.

Equations: May you give a brief description of the variables and parameters next to the equations.

Please insert a figure about ε-NTU in dependance on the solutions and setup conditions. It woud be helpful to understand your conclusions better.

References:
Would it be useful to make a comparison with data of E. Kozubal's et al's report on Low-Flow Liquid Desiccant Air-Conditioning: Demonstrated Performance and Cost Implications, https://www.nrel.gov/docs/fy14osti/60695.pdf ?

Author Response

(The authors gave the same response as above.)
